# Deep Model Transferability from Attribution Maps

**Jie Song**[1,3], **Yixin Chen**[1], **Xinchao Wang**[2], **Chengchao Shen**[1], **Mingli Song**[1,3]
[1]Zhejiang University, [2]Stevens Institute of Technology
[3]Alibaba-Zhejiang University Joint Institute of Frontier Technologies
{sjie,chenyix,chengchaoshen,brooksong}@zju.edu.cn
xinchao.wang@stevens.edu

## Abstract

Exploring the transferability between heterogeneous tasks sheds light on their intrinsic interconnections, and consequently enables knowledge transfer from one task to another so as to reduce the training effort of the latter. In this paper, we propose an embarrassingly simple yet very efficacious approach to estimating the transferability of deep networks, especially those handling vision tasks. Unlike the seminal work of *taskonomy* that relies on a large number of annotations as supervision and is thus computationally cumbersome, the proposed approach requires no human annotations and imposes no constraints on the architectures of the networks. This is achieved, specifically, via projecting deep networks into a *model space*, wherein each network is treated as a point and the distances between two points are measured by deviations of their produced attribution maps. The proposed approach is several-magnitude times faster than taskonomy, and meanwhile preserves a task-wise topological structure highly similar to the one obtained by taskonomy. Code is available at `https://github.com/zju-vipa/TransferbilityFromAttributionMaps`.

## 1 Introduction

Deep learning has brought about unprecedented advances in many if not all the major artificial intelligence tasks, especially computer vision ones. The state-of-the-art performances, however, come at the costs of the often burdensome training process that requires an enormous number of human annotations and GPU hours, as well as the partially interpretable and thus the only intermittently predictable black-box behaviors. Understanding the intrinsic relationships between such deep-learning tasks, if any, may on the one hand elucidate the rationale of the encouraging results achieved by deep learning, and on the other hand allows for more predictable and explainable transfer learning from one task to another, so that the training effort can be significantly reduced.

The seminal work of *taskonomy* [37] made the pioneering attempt towards disentangling the relationships between visual tasks through a computational approach. This is accomplished by training first all the task models and then all the feasible transfers among models, in a fully supervised manner. Based on the obtained transfer performances, an affinity matrix of transferability is derived, upon which an Integer Program can be further imposed to compute the final budget-constrained task-transferability graph. Despite the intriguing results achieved, the training cost, especially that for the combinatorial-based transferability learning, makes taskonomy prohibitively expensive to estimate. Even for the first-order transferability estimation, the training costs grow quadratically with respect to the number of tasks involved; when adding a new task to the graph, the transferability has to be explicitly trained between the new task and all those in the task dictionary.

In this paper, we propose an embarrassingly simple yet competent approach to estimating the transferability between different tasks, with a focus on the computer vision ones. Unlike taskonomy that relies on training the task-specific models and their transferability using human annotations, in

our approach we assume no labelled data are available, and we are given only the pre-trained deep networks, which can be nowadays found effortless online. Moreover, we do not impose constraints on the architectures of the deep networks, such as networks handling different tasks sharing the same architectures.

At the heart of our approach is to project pre-trained deep networks into a common space, termed *model space*. The model space accepts networks of heterogeneous architectures and handling different tasks, and transforms each network into a point. The distance between two points in the model space is then taken to be the measure of their relatedness and the consequent transferability. Such construction of the model space enables prompt model insertion or deletion, as updating the transferability graph boils down to computing nearest neighbors in the model space, which is therefore much lighter than taskonomy that requires the pair-wise re-training for each newly added task.

The projection to the model space is attained by feeding unlabelled images, which can be obtained handily online, into a network and then computing the corresponding *attribution maps*. An attribution map signals pixels in the input image highly relevant to the downstream tasks or hidden representations, and therefore highlights the "attention" of a network over a specific task. In other words, the model space can be thought as a space defined on top of attribution maps, where the affinity between points or networks is evaluated using the distance between their produced attribution maps, which again, requires no supervision and can be computed really fast.

The intuition behind adopting attribution maps for network-affinity estimation is rather straight-forward: models focusing on similar regions of input images are expected to produce correlated representations, and thus potentially give rise to favorable transfer-learning results. This assumption is inspired by the work of [36], which utilizes the attention of a teacher model to guide the learning of a student and produces encouraging results. Despite its very simple nature, the proposed approach yields truly promising results: it leads to a speedup factor of several magnitudes of times and mean-while maintains a highly similar transferability topology, as compared to taskonomy. In addition, experiments on vision tasks beyond those involved in taskonomy also produce intuitively plausible results, validating the proposed approach and providing us with insights on their transferability.

Our contribution is therefore a lightweight and effective approach towards estimating transferability between deep visual models, achieved via projecting each model into a common space and approximating their affinity using attribution maps. It requires no human annotations and is readily applicable to pre-trained networks specializing in various tasks and of heterogeneous architectures. Running at a speed several magnitudes faster than taskonomy and producing competitively similar results, the proposed model may serve as a competent transferability estimator and an effectual substitute for taskonomy, especially when human annotations are unavailable, when the model library is large in size, or when frequent model insertion or update takes place.

## 2  Related Work

We briefly review here some topics that are most related to the proposed work, including model reusing, transfer learning, and attribution methods for deep models.

**Model Reusing.**   Reusing pre-trained models has been an active research topic in recent years. Hinton *et al.* [9] firstly propose the concept of "knowledge distillation" where the trained cumbersome teacher models are reused to produce soft labels for training a lightweight student model. Following their teacher-student scheme, some more advanced methods [24, 36, 6, 15] are proposed to fully exploit the knowledge encoded in the trained teacher model. However, in these works all the teachers and the student are trained for the same task. To reuse models of different tasks, Rusu *et al.* [25] propose the progressive neural net to extract useful features from multiple teachers for a new task. Parisotto *et al.* [19] propose "Actor-Mimic" to use the guidance from several expert teachers of distinct tasks. However, none of these works explore the relatedness among different tasks. In this paper, by explicitly modeling the model transferability, we provide an effective method to pick a trained model most beneficial for solving the target task.

**Transfer Learning.**   Another way of reusing trained models is to transfer the trained model to another task by reusing the features extracted from certain layers. Razavian *et al.* [22] demonstrated that features extracted from deep neural networks could be used as generic image representations to

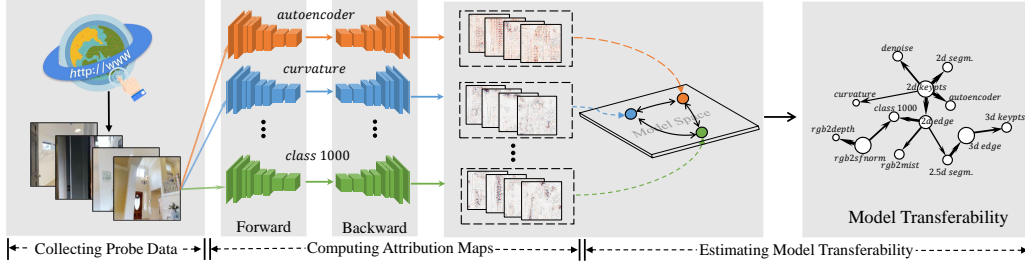

Figure 1: An illustrative diagram of the workflow of the proposed method. It mainly consists of three steps: collecting probe data, computing attribution maps, and estimating model transferability.

tackle the diverse range of visual tasks. Yosinski *et al.* [34] investigated the transferability of deep features extracted from every layer of a deep neural network. Azizpour *et al.* [2] investigated several factors affecting the transferability of deep features. Recently, the effects of pre-training datasets for transfer learning are studied [13, 7, 12, 33, 23]. None of these works, however, explicitly quantify the relatedness among different tasks or trained models to provide a principled way for model selection. Zamir *et al.* [37] proposed a fully computational approach, known as taskonomy, to address this challenging problem. However, taskonomy requires labeled data and is computationally expensive, which limited its applications in large-scale real-world problems. Recently, Dwivedi and Roig [4] proposed to use representation similarity analysis to approximate the task taxonomy. In this paper, we introduce a model space for modeling task transferability and propose to measure the transferability via attribution maps, which, unlike taskonomy, requires no human annotations and works directly on pre-trained models. We believe our method is a good complement to existing works.

**Attribution Methods for Deep Models.** Attribution refers to assigning importance scores to the inputs for a specified output. Existing attribution methods can be mainly divided into two groups, including perturbation- [38, 39, 40] and gradient-based [28, 3, 27, 30, 26, 18, 1] methods. Perturbation-based methods compute the attribution of an input feature by making perturbations, *e.g.*, removing, masking or altering, to individual inputs or neurons and observe the impact on later neurons. However, such methods are computationally inefficient as each perturbation requires a separate forward propagation through the network. Gradient-based methods, on the other hand, estimate the attributions for all input features in one or few forward and backward passes throughout the network, which renders them generally more efficient. Simonyan *et al.* [28] construct attributions by taking the absolute value of the partial derivative of the target output with respect to the input features. Later, Layer-wise Relevance Propagation ($\epsilon$-LRP) [3], gradient*input [27], integrated gradients [30] and deepLIFT [26] are proposed to aid understanding the information flow of deep neural networks. In this paper, we directly adopt some of these off-the-shelf methods to produce the attribution maps. Devising more suitable attribution method for our problem is left to future work.

## 3 Estimating Model Transferability from Attribution Maps

We provide in this section the details of the proposed transferability estimator. We start by giving the problem setup and an overview of the method, followed by describing its three steps, and finally show the efficiency analysis.

### 3.1 Problem Setup

Assume we are given a set of pre-trained deep models $\mathcal{M} = \{m_1, m_2, ..., m_N\}$, where $N$ is the total number of models involved. No constraints are imposed on the architectures of these models. We use $t_i$ to denote the task handled by model $m_i$, and use $\mathcal{T} = \{t_1, t_2, ..., t_N\}$ to denote the *task dictionary*, *i.e.*, the set of all the tasks involved in $\mathcal{M}$. Furthermore, we assume that no labeled annotations are available. Our goal is to efficiently quantify the transferability between different tasks in $\mathcal{T}$, so that given a target task, we can read out from the learned transferability matrix the source task that potentially yields the highest transfer performance.

## 3.2 Overview

The core idea of our method is to embed the pre-trained deep models into the model space, wherein models are represented by points and model transferability is measured by the distance between corresponding points. To this end, we utilize the attribution maps to construct such a model space. The assumption is that related models should produce similar attribution maps for the same input image. The workflow of our method consists of three steps, as shown in Figure 1. First, we collect an unlabeled *probe dataset*, which will be used to construct the model space, from a randomly selected data distribution. Second, for each trained model, we adopt off-the-shelf attribution methods to compute the attribution maps of all images in the constructed probe dataset. Finally, for each model, all its attribution maps are collectively viewed as a single point in the model space, based on which the model transferability is estimated. In what follows, we provide details for each of the three steps.

## 3.3 Key Steps

**Step 1: Building the Probe Dataset.** As deep models handling different tasks or even the same one may be of heterogeneous architectures or trained on data from various domains, it is non-trivial to measure their transferability directly from their outputs or intermediate features. To bypass this problem, we feed the same input images to these models and measure the model transferability by the similarity of their response to the same stimuli. We term the set of all such input images *probe data*, which is shared by all the tasks involved.

Intuitively, the probe dataset should be designed not only large in size but also rich in diversity, as models in $\mathcal{M}$ may be trained on various domains for different tasks. However, experiments show that the proposed method works surprisingly well even when the probe data are collected in a single domain and of moderately small size ($\sim 1,000$ images). The produced transferability relationship is highly similar to the one derived by taskonomy. This property renders the proposed method attractive as little effort is required for collecting the probe data. More details can be found in Section 4.2.3.

**Step 2: Computing Attribution Maps.** Let us denote the collected probe data by $\mathcal{X} = \{X_1, X_2, ..., X_{N_p}\}$, $X_i = [x_1^i, x_2^i, ..., x_{WHC}^i] \in \mathbb{R}^{WHC}$, where $W$, $H$ and $C$ respectively denote the width, the height and the channels of the input images, and $N_p$ is the size of the probe data. Note that for brevity the maps are symbolized in vectorization form here. For model $m_i$, it takes an input $\tilde{X} = T_i(X) \in \mathbb{R}^{W_i H_i C_i}$ and produces a hidden representation $R = [r_1, r_2, ..., r_D]$. Here, $T_i$ serves as a preprocessing function that transforms the images in probe data for model $m_i$, as we allow different models to take images of different sizes as input, and $D$ is the dimension of the representation. For each model $m_i$ in $\mathcal{M}$, our goal in this step is to produce an attribution map $A_j^i = [a_{j1}^i, a_{j2}^i, ...] \in \mathbb{R}^{WHC}$ for each image $X_j$ in the probe data $\mathcal{X}$.

In fact, an attribution map $A_j^{i,k}$ can be computed for each unit $r_k$ in $R$. However, as we consider the transferability of $R$, we average the attribution maps of all $r$ in $R$ as the overall attribution map of $R$. Formally, we have $A_j^i = \frac{1}{D} \sum_{k=1}^{D} A_j^{i,k}$. Specifically, here we adopt three off-the-shelf attribution methods to produce the attribution maps: saliency map [28], gradient * input [27], and $\epsilon$-LRP [3]. Saliency map computes attributions by taking the absolute value of the partial derivative of the target output with respect to the input. Gradient * input refers to a first-order taylor approximation of how the output would change if the input was set to zero. $\epsilon$-LRP, on the other hand, computes the attributions by redistributing the prediction score (output) layer by layer until the input layer is reached. For all the three attribution methods, the overall attribution map $A_j^i$ can be computed through one single forward-and-backward propagation [1] in Tensorflow. The formulations of the three attribution maps are summarized in Table 1. More details can be found from [28, 27, 3, 1].

Table 1: Mathematical formulations of saliency map [28], gradient * input [27] and $\epsilon$-LRP [3]. Note that the superscript $g$ denotes a novel definition of partial derivative [1].

| Method | Saliency Map [28] | Gradient * Input [27] | $\epsilon$-LRP [3, 1] |
|---|---|---|---|
| $\tilde{A}_j^{i,k}$ | $\left[ \left\| \frac{\partial r_k}{\partial x_d^j} \right\| \right]_{d=1}^{W_i H_i C_i}$ | $\left[ x_d^j \cdot \frac{\partial r_k}{\partial x_d^j} \right]_{d=1}^{W_i H_i C_i}$ | $\left[ x_d^j \cdot \frac{\partial^g r_k}{\partial x_d^j} \right]_{d=1}^{W_i H_i C_i}, g = \frac{f(z)}{z}$ |

For model $m_i$, the produced attribution map $\tilde{A}^i$ is of the same size as the input $\tilde{X}$, i.e., $\tilde{A}^i \in \mathbb{R}^{W_i H_i C_i}$. We do the inverse of $T$ to transform the attribution maps back to the same size as the images in the probe data: $A^i = T^{-1}(\tilde{A}^i)$, $A^i \in \mathbb{R}^{WHC}$. As attribution maps of all models are transformed into the same size, the transferability can be computed based on these maps.

**Step 3: Estimating Model Transferability.** Once step 2 is completed, we have $N_p$ attribution maps $\mathcal{A}^i = \{A_1^i, A_2^i, ..., A_{N_p}^i\}$ for each model $m_i$, where $A_j^i$ denotes the attribution map of $j$-th image $X_j$ in $\mathcal{X}$. The model $m_i$ can be viewed as a sample in the model space $\mathbb{R}^{NWHC}$, formed by concatenating all the attribution maps. The distance of two models are taken to be

$$d(m_i, m_j) = \frac{N_p}{\sum_{k=1}^{N_p} cos\_sim(A_k^i, A_k^j)}, \tag{1}$$

where $cos\_sim(A_k^i, A_k^j) = \frac{A_k^i \cdot A_k^j}{\|A_k^i\| \cdot \|A_k^j\|}$. The model transferability map, which measures the pairwise transferability relationships, can then be derived based on these distances. The model transferability, as shown by taskonomy [37], is inherently asymmetric. In other words, if model $m_i$ ranks first in being transferred to task $t_j$ among all the models (except $m_j$) in $\mathcal{M}$, $m_j$ does not necessarily rank first in being transferred to task $t_i$. Yet, the proposed model space is symmetric in distance, as we have $d(m_i, m_j) = d(m_j, m_i)$. We argue that the symmetric property of the distance in the model space makes little negative effect on the transferability relationships, as the task transferability rankings of the source tasks are computed by relative comparison of distances. Experiments demonstrate that with the symmetric model space, the proposed method is able to effectively approximate the asymmetric transferability relationships produced by taskonomy.

### 3.4 Efficiency Analysis

Here we make a rough comparison between the efficiency of the proposed approach and that of taskonomy. As we assume task-specific trained models are available, we compare the computation cost of our method with that of only the transfer modeling in taskonomy. For taskonomy, let us assume the transfer model is trained for $E$ epochs on the training data of size $N$, then for a task dictionary of size $T$, the computation cost can be approximately denoted as $ENT(T-1)$-times forward-and-backward propagation[1]. For our method working on the probe dataset, however, only one time of forward-and-backward propagation is required. The overall computation cost for building the model space in our method is about $TM$-times forward-and-backward propagation, where $M$ is the size of the probe dataset and usually $M \ll N$. The proposed method is thus about $\frac{EN(T-1)}{M}$-times more efficient than taskonomy. This also means the speedup over taskonomy will be even more significant, if more tasks are involved and hence $T$ enlarges.

In our experiments, the proposed method takes about 20 GPU hours to compute the pairwise transferability relationships on one Quadro P5000 card for 20 pre-trained taskonomy models, while taskonomy takes thousands of GPU hours on the cloud[2] for the same number of tasks.

## 4 Experiments

### 4.1 Experimental Settings

**Pre-trained Models.** Two groups of trained models are adopted to validate the proposed method. In the first group, we adopt 20 trained models of single-image tasks released by taskonomy [37], of which the task relatedness has been constructed and also released. It is used as the oracle to evaluate the proposed method. Note that all these models adopt an encoder-decoder architecture, where the encoder is used to extract representations and the decoder makes task predictions. For these models, the attribution maps are computed with respect to the output of the encoder.

To further validate the proposed method, we construct a second group of trained models which are collected online. We have managed to obtain 18 trained models in this group: two VGGs [29] (VGG16, VGG19), three ResNets [8] (ResNet50, ResNet101, ResNet152), two Inceptions (Inception V3 [32],

| **Input** | Curvature | Denoise | Edge 2D | Edge 3D | Keypoint 2D | Keypoint 3D | Colorization | Reshade |
|---|---|---|---|---|---|---|---|---|
| 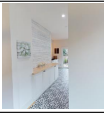 |  |  |  |  |  |  |  |  |
|  |  |  |  |  |  |  |  |  |
| **Input** | Rgb2depth | Rgb2mist | Rgb2sfnorm | Segment 2D | Vanishing Pts | Seg Semantic | Class 1000 | Class Places |
|  |  |  |  |  |  |  |  |  |
|  |  |  |  |  |  |  |  |  |

Figure 2: Visualization of attribution maps produced using $\epsilon$-LRP on taskonomy models. Some tasks produce visually similar attribution maps, such as Rgb2depth and Rgb2mnist.

Inception ResNet V2 [31]), three MobileNets [10] (MobileNet, 0.5 MobileNet, 0.25 MobileNet), four Inpaintings [35] (ImageNet, CelebA, CelebA-HQ, Places), FCRN [14], FCN [17], PRN [5] and Tiny Face Detector [11]. All these models are also viewed in an encoder-decoder architecture. The sub-model which produces the most compact features is viewed as the encoder and the remainder as the decoder. Similar to taskonomy models, the attribution maps are computed with respect to the output of the encoder. More details of these models can be found in the supplementary material.

**Probe Datasets.** We build three datasets, taskonomy data [37], indoor scene [20], and COCO [16], as the probe data to evaluate our method. The domain difference between taskonomy data and COCO is much larger than that between taskonomy data and indoor scene. For all the three datasets, we randomly select about $1,000$ images to construct the probe datasets. More details of the three probe datasets are provided in the supplementary material. In Section 4.2.3, we demonstrate the performances of the proposed method evaluated on these three probe datasets.

## 4.2 Experiments on Models in Taskonomy

### 4.2.1 Visualization of Attribution Maps

We first visualize the attribution maps produced by various trained models for the same input images. Two examples are given in Figure 2. Attribution maps are produced by $\epsilon$-LRP on taskonomy data. From the two examples, we can see that some tasks produce visually similar attribution maps. For example, ⟨Rgb2depth, Rgb2mist⟩[3], ⟨Class 1000, Class Places⟩ and ⟨Denoise, Keypoint 2D⟩. In each cluster, trained models pay their "attentions" to the similar regions, thus the "knowledge" they learned are intuitively highly correlated (as seen in Section 4.2.2) and can be transferred to each other (as seen in Section 4.2.3). Two examples may produce conclusions where the constructed model transferability deviates from the underlying model relatedness. However, such deviation is alleviated by aggregating the results of more examples drawn from the data distribution. For more visualization examples, please see the supplementary material.

### 4.2.2 Rationality of the Assumption

Here we adopt Singular Vector Canonical Correlation Analysis (SVCCA) [21] to validate the rationality underlying our assumption: if tasks produce similar attribution maps, the representations extracted from corresponding models should be highly correlated, thus they are expected to yield favorable transfer-learning performance to each other. In SVCCA, each neuron is represented by an *activation vector*: its set of response to a set of inputs and hence the layer can be represented by the subspace

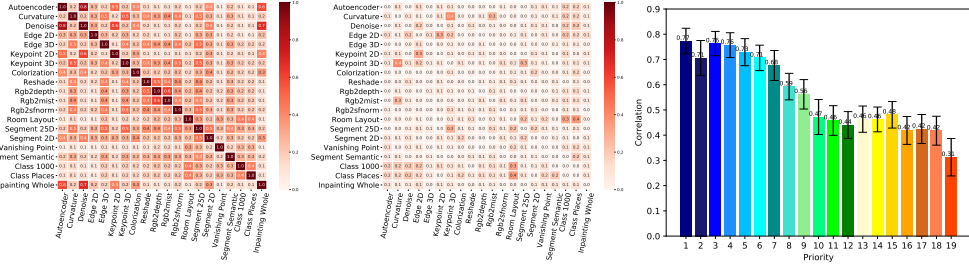

Figure 3: **Left**: visualization of the correlation matrix from SVCCA. **Middle**: the difference between correlation matrix from SVCCA and the transferability matrix derived from attribution maps. Both of them are normalized for better visualization. **Right**: the Correlation-Priority Curve (CPC).

spanned by the activation vectors of all the neurons in this layer. SVCCA first adopts Singular Value Decomposition (SVD) of each subspace to obtain new subspaces that comprise the most important directions of the original subspaces, and then uses Canonical Correlation Analysis (CCA) to compute a series of correlation coefficients between the new subspaces. The overall correlation is measured by the average of these correlation coefficients.

Experimental results on taskonomy data with $\epsilon$-LRP are shown in Figure 3. In the left, the correlation matrix over the pre-trained taskonomy models is visualized. In the middle, we plot the difference between the correlation matrix and the model transferability matrix derived from attribution maps in the proposed method. It can be seen that the values in the difference matrix are in general small, implying that the correlation matrix is highly similar to the model transferability matrix. To further quantify the overall similarity between these two matrices, we compute their Pearson correlation ($\rho_p = 0.939$) and Spearman correlation ($\rho_s = 0.660$). All these results show that the similarity of attribution maps is a good indicator of the correlation between representations.

In addition, we can see that some tasks, like Edge3d and Colorization, tend to be more correlated to other tasks, as the colors of the corresponding row or column are darker than those of others, while some other tasks are not, like Vanishing Point. In taskonomy, the priorities[4] of Edge3d, Colorization and Vanishing Point are $5.4$, $5.8$ and $14.2$, respectively. It indicates that more correlated representations tend to be more suitable for transferring learning to each other. To make this clearer, we depict the Correlation-Priority Curve (CPC) in the right of Figure 3. In this figure, for each priority $p$ shown on the abscissa, the correlation shown on the ordinate is computed as $correlation(p) = \frac{1}{N} \sum_{i \neq j} \mathbb{I}(r_j^i = p) \rho_{i,j}$, where $\mathbb{I}$ is the indicator function and $\rho_{i,j}$ is the correlation between representations extracted from two models $m_i$ and $m_j$. It can be seen that as the priority becomes lower, the average correlation becomes weaker. All these results verify the rationality underlying the assumption.

### 4.2.3 Deep Model Transferability

We adopt two evaluation metrics, P@K and R@K[5], which are widely used in the information retrieval field, to compare the model transferability constructed from our method with that from tasknomy. Each target task is viewed as a query, and its top-5 source tasks that produce the best transferring performances in taskonomy are regarded as relevant to the query. To better understand the results, we introduce one baseline using random ranking, and the oracle, the ideal method which always produces the perfect results. Additionally, we also evaluate SVCCA for computing the model transferability relationships. The experimental results are depicted in Figure 4. Based on the results, we can make the following conclusions.

- The topology structure of the model transferability derived from the proposed method is similar to that of oracle. For example, when only top-3 predictions are examined, the precision can be about $85\%$ on COCO with $\epsilon$-LRP. To see this clearer, we also depict the task similarity tree constructed

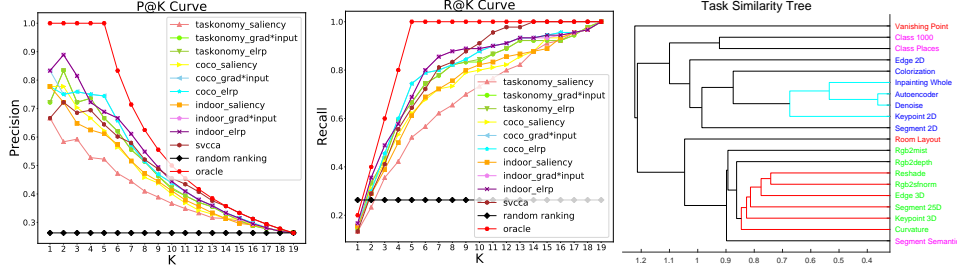

Figure 4: From left to right: P@K curve, R@K curve and task similarity tree constructed by $\epsilon$-LRP. Results of SVCCA are produced using validation data from taskonomy.

by agglomerative hierarchical clustering in Figure 4. This tree is again highly similar to that of taskonomy where 3D, 2D, geometric, and semantic tasks cluster together.

- $\epsilon$-LRP and gradient* input generally produce better performance than saliency. This phenomenon can be in part explained by the fact that saliency generates attributions entirely based on gradients that denote the direction for optimization. However, the gradients are not able to fully reflect the relevance between the inputs and the outputs of the deep model, thus leading to inferior results. It also implies the attribution method can affect the performance of our method. Devising better attribution methods may further promote the accuracy of our method, which is left as future work.

- The proposed method works quite well on the probe data from different domains, such as indoor scene and COCO. It implies that the proposed method is robust to different choices of the probe data to some degree, which makes the data collection effortless. Furthermore, it can be seen that the probe data from indoor scene and COCO surprisingly better predict the taskonomy transferability than the probe data from taskonomy data. We conjecture that more complex textures disentangle the attributions better, thus the probe data from COCO and indoor scene which are generally more complex in texture yield superior results to taskonomy as probe data. However, more research is necessary to discover if the explanation holds in general.

- SVCCA also works well in estimating the transferability of taskonomy models. However, the proposed method yields superior or comparable performance to SVCCA when using gradient * input and $\epsilon$-LRP for attribution. What's more, as the proposed method measures transferability by computing distances, it is several times more efficient than SVCCA, especially when the hidden representation is large in dimension or a new task is added into a large task dictionary.

With all these observations and the fact that the proposed method is significantly more efficient than taskonomy, the proposed method is indeed an effectual substitute for taskonomy, especially when human annotations are unavailable, when the model library is large in size, or when frequent model insertion and update takes place.

## 4.3 Experiments on Models beyond Taskonomy

To give a more comprehensive view of the proposed method, we also conduct experiments on the online collected pre-trained models beyond taskonomy. Results are shown in Figure 5. The left two subfigures show the correlation matrix from SVCCA and the model transferability matrix produced by our method. The right two subfigures depict the task similarity trees produced by SVCCA and the proposed method. The classification and inpainting models are listed in different colors. We have the following observations.

- The proposed method produces an affinity matrix and a task similarity tree alike those derived from SVCCA, although the collected models are heterogeneous in architectures, tasks, and input size. These results further validate that models producing similar attribution maps also produce highly correlated representations.

- All the ImageNet-trained classification models, despite their different architectures, tend to cluster together. Furthermore, the same-task trained models with the similar architectures tend to be more related than with dissimilar architectures. For example, ResNet50 is more related to ResNet101 and ResNet152 than VGG, MobileNet and Inception models, indicating that the architecture plays a certain role in regularization for solving the tasks.

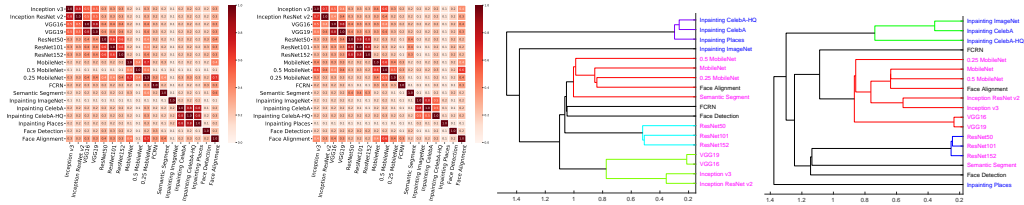

Figure 5: Results on collected models beyond taskonomy. From left to right: affinity matrix from SVCCA, affinity matrix from attribution maps, task similarity tree from SVCCA, and task similarity tree from attribution maps.

- The inpainting models, albeit trained on data from different data domain, also tend to cluster together. It implies that different models of the same task, albeit trained on data from different data domain, tend to play similar role in transfer learning. However, more research is necessary to verify if this observation holds in general.

We also merge the two groups into one to further evaluate the proposed method, of which the results are provided in the supplementary material, providing us with more insights on model transferability.

## 5 Conclusion

We introduce in this paper an embarrassingly simple yet efficacious approach towards estimating the transferability between deep models, without using any human annotation. Specifically, we project the pre-trained models of interest into a model space, wherein each model is treated as a point and the distance between two points are used to approximate their transferability. The projection to the model space is achieved by computing the attribution maps from the unlabelled probe dataset. The proposed approach imposes no constraints on the architectures on the models, and turns out to be robust to the selection of the probe data. Despite the lightweight construction, it yields a transferability map highly similar to the one obtained by taskonomy yet runs at a speed several magnitudes faster, and therefore may serve as a compact and express transferability estimation, especially when no annotations are available, the model library is large in size, or frequent model insertion or update takes place.

### Acknowledgments

This work is supported by National Key Research and Development Program (2016YFB1200203), National Natural Science Foundation of China (61572428), Key Research and Development Program of Zhejiang Province (2018C01004), and the Major Scientifc Research Project of Zhejiang Lab (No. 2019KD0AC01).

## Footnotes

[1]Here for simplicity, we ignore the computation-cost difference caused by the model architectures.

[2]As the hardware configurations are not clear here, we list the GPU hours only for perceptual comparison.

[3]Here we use ⟨⟩ to denote a cluster of tasks, of which the attribution maps are highly similar.

[4]The priority of a task $i$ refers to the average ranking when transferred to other tasks: $p_i = \frac{1}{N} \sum_j^N r_j^i$, where $r_j^i$ denotes the ranking of task $i$ when transferred to task $j$. A smaller value of $p$ denotes a higher priority.

[5]P: precision, R: recall, @K: only the top-K results are examined.

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
