[Supplementary Material · NeurIPS-3328-SUPP-FINAL.pdf]

# Deep Model Transferability from Attribution Maps
## – *Supplementary Material* –

**Jie Song**[1,3], **Yixin Chen**[1], **Xinchao Wang**[2], **Chengchao Shen**[1], **Mingli Song**[1,3]
[1]Zhejiang University, [2]Stevens Institute of Technology
[3]Alibaba-Zhejiang University Joint Institute of Frontier Technologies
{sjie,chenyix,chengchaoshen,brooksong}@zju.edu.cn
xinchao.wang@stevens.edu

## 1 Pre-trained Models

We adopt two groups of pre-trained models to validate the proposed method. The first group consists of the pre-trained taskonomy models [24]. All the taskonomy models adopt an encoder-decoder architecture. For all tasks, the encoders are identical and implemented by the modified ResNet-50. The architectures of the decoders depend on the tasks. We adopt 20 trained models of single-image tasks released by taskonomy[1]: Autoencoder, Curvature, Denoise, Edge 2D, Edge 3D, keypoint 2D, Keypoint 3D, Colorization, Reshade, Rgb2depth, Rgb2minst, Rgb2sfnorm, RoomLayout, Segment 25D, Segment 2D, VanishingPoint, SegmentSemantic, Class 1000, Class Places and Inpainting Whole. Please refer to [24] for more details about these models.

Table 1: Details of the collected models. "Layer Name" refers to the layers w.r.t. which the attribution maps are computed. "Pre-Logits" indicates the layer previous to the one which produces logits.

| Model | Task | Training Data | Input Size | Pre-trained | Layer Name |
|---|---|---|---|---|---|
| Inception V3 [21] | Classification | ILSVRC2012 [14] | $299 \times 299 \times 3$ | ✗ | Pre-Logits |
| Inception-ResNet V2 [20] | Classification | ILSVRC2012 [14] | $299 \times 299 \times 3$ | ✗ | Pre-Logits |
| VGG 16 [19] | Classification | ILSVRC2012 [14] | $224 \times 224 \times 3$ | ✗ | Pre-Logits |
| VGG 19 [19] | Classification | ILSVRC2012 [14] | $224 \times 224 \times 3$ | ✗ | Pre-Logits |
| ResNet V1 50 [4] | Classification | ILSVRC2012 [14] | $224 \times 224 \times 3$ | ✗ | Pre-Logits |
| ResNet V1 101 [4] | Classification | ILSVRC2012 [14] | $224 \times 224 \times 3$ | ✗ | Pre-Logits |
| ResNet V1 152 [4] | Classification | ILSVRC2012 [14] | $224 \times 224 \times 3$ | ✗ | Pre-Logits |
| Mobilenet V1 [5] | Classification | ILSVRC2012 [14] | $224 \times 224 \times 3$ | ✗ | Pre-Logits |
| 50% Mobilenet V1 [5] | Classification | ILSVRC2012 [14] | $160 \times 160 \times 3$ | ✗ | Pre-Logits |
| 25% Mobilenet V1 [5] | Classification | ILSVRC2012 [14] | $128 \times 128 \times 3$ | ✗ | Pre-Logits |
| Generative Inpainting [23] | Inpainting | Places2 [25] | $512 \times 680 \times 3$ | ✗ | "allconv12"† |
| Generative Inpainting [23] | Inpainting | CelebA [10] | $256 \times 256 \times 3$ | ✗ | "allconv12"† |
| Generative Inpainting [23] | Inpainting | CelebA-HQ [7] | $256 \times 256 \times 3$ | ✗ | "allconv12"† |
| Generative Inpainting [23] | Inpainting | ILSVRC2012 [14] | $256 \times 256 \times 3$ | ✗ | "allconv12"† |
| FCRN [8] | Depth Estimation | NYU Depth v2 [17] | $512 \times 512 \times 3$ | ✓ | "layer1"† |
| PRN [3] | Face Alignment | 300W-LP [26] | $256 \times 256 \times 3$ | ✗ | "ResBlock10"† |
| FCN [11] | Semantic Segmentation | PASCAL VOC [2] | $512 \times 512 \times 3$ | ✓ | "Conv8"† |
| Tiny Face Detector [6] | Face Detection | WIDER FACE [22] | $512 \times 512 \times 3$ | ✓ | "res4b"† |

† : the name given in the source code.

To further validate our method, we also collect another 18 pre-trained models online beyond those involved in taskonomy, including two VGGs [19] (VGG16, VGG19), three ResNets [4] (ResNet50, ResNet101, ResNet152), two Inceptions (Inception V3 [21], Inception ResNet V2 [20]), three MobileNets [5] (MobileNet, 0.5 MobileNet, 0.25 MobileNet), four Inpaintings [23] (ImageNet,

Figure S1: Visualization of attribution maps produced by saliency [18] and gradient*input [15].

CelebA, CelebA-HQ, Places), FCRN [8], FCN [11], PRN [3] and Tiny Face Detector [6]. Details of these models are summarized in Table 1. These models can be further categorized into three groups:

- Classification models: trained on the same data (ILSVRC2012 [14]), for the same task (1000-way classification), but in different model architectures (Inception, ResNet, VGG, MobileNet). We adopt the pre-trained models released by Tensorflow-Slim[2] [16] lib.
- Inpainting models: in the same model architecture, trained for the same task, but on different datasets (Places2 [25], CelebA [10], CelebA-HQ [7] and ILSVRC2012 [14]). We adopt the pre-trained models released by [23][3].
- Other models: models in this group are heterogeneous in architectures, tasks and training data. This group consists of four models, including FCRN [8], FCN [11], PRN [3] and Tiny Face Detector [6]. Pre-trained models can be found in their project pages.

In our experiments, we also merge the two groups of models to form a more comprehensive group. Experiments on this new group will provide us more insights into the proposed method.

## 2 Probe Datasets

Here we provide more details about the three probe datasets used in the proposed method.

Figure S2: Visualization of transferability matrices produced by saliency [18], gradient*input [15] and $\epsilon$-LRP [1] on images randomly selected from taxonomy data [24], indoor scene [13], COCO data [9].

**Taskonomy data** On taskonomy data [24], we construct the probe data by selecting images from the validation data of the "Tiny" partition. In the validation set of Tiny partition, images are collected from 5 different buildings. We randomly select 200 images from each of these 5 buildings, constructing a probe dataset consisting of 1,000 images.

**Indoor Scene** Indoor Scene [13] is a dataset used for indoor scene recognition. The original database contains 67 indoor categories, and a total of 15,620 images. We randomly select 15 images from each of these 67 categories, constructing a probe dataset consisting of 1,005 images.

**COCO** The COCO [9] dataset is designed for several purpose such as detection, caption and so on. On this dataset, we randomly select 1,000 images from the 2014 Val dataset to construct the probe dataset for evaluating the proposed method.

The styles of images in these three datasets are very different. Generally speaking, the textures of images in taskonomy data are simple. However, the textures of images in Indoor Scene and COCO are relatively more complex.

## 3 Visualization of Attribution Maps

In this section, we visualize attribution maps of examples from taskonomy data [24] for better understanding of our method. Here attribution maps are produced by saliency maps [18] and gradient * input [15]. Results are visualized in Figure S1. It can be seen that some tasks tend to produce much more similar attribution maps than others. For example, <Rgb2depth, Rgb2mist> and <Class 1000, Class Places>. These producing-similar-attribution tasks are proved to be highly related in the

Figure S3: Task similar trees produced by our method with saliency [18], gradient*input [15] and $\epsilon$-LRP [1] on images randomly selected from taxonomy data [24], indoor scene [13], COCO data [9].

task structure found in taxonomy and thus producing favorable transfer performance to each other. Some examples may produce misleading results. However, the conclusions made by statistically aggregating the results of all the randomly sampled examples become more reliable.

## 4    Visualization of Deep Model Transferability

In this section, we provide the results of deep model transferability found by the proposed method. Here we show the model transferability in two way: visualization of the affinity matrix and the task similarity tree. The visualization of attribution maps is provided in Figure S2. In this figure, the affinity matrices in each row are produced by the same attribution method which is listed on the left. The probe data used for matrices in each column are listed on the top. The results demonstrate that:

- For each attribution method, no matter which probe data is adopted, the produced affinity matrixes (in the same row) are highly similar. It implies that the proposed method is insensitive to the choice of probe dataset to some degree, which renders our method robust and flexible.

- On each probe dataset, the transferability matrices produced by $\epsilon$-LRP are visually similar to gradient * input. However, the saliency seems to produce visually more dissimilar results. Actually, the transferability matrices produced by $\epsilon$-LRP and gradient*input are more consistent with that produced by taskonomy than that produced by saliency. The fact accounting for this phenomenon may be that saliency generate attributions entirely based on gradients which denote the direction for optimization. However, the gradients can't fully reflect the relevance between the inputs and the outputs of the deep model, thus leading to inferior results in our method.

To better understand the results produced by different attribution methods and probe data, we construct the task similar trees by agglomerative hierarchical clustering. The constructed task similar trees are depicted in Figure S3. To be consistent with taskonomy [24], 3D, 2D, geometric and semantic tasks are listed in different fonts. Taskonomy has shown that the tasks in the same font plays similar roles

Figure S4: Model transferability matrices and the task similar trees produced by our method with $\epsilon$-LRP on various size (listed below the subfigures) of probe data randomly selected from COCO.

in transferring to other tasks, thus they should be clustered together. The results shown in Figure S3 indicate that:

- For each attribution method, the task similar trees obtained on the three probe datasets are highly similar. It again verifies that our method is insensitive to the choice of the probe data.

- For gradient * input [15] and $\epsilon$-LRP [1], the tasks of the same type are grouped into the same cluster with few exceptions. These results are highly similar to that produced by taskonomy. Considering that our method is much more computation efficient, we argue that our method is much more scalable and practical.

- If we adopt saliency for attribution in our method, the constructed similar trees are a little different, where the 2D tasks tend to be clustered into two groups: <Inpainting, Edge2D, Colorization> and <Keypoint2D, Autoencoder, Segment2D>.

We also investigate how the size of the probe data affects the results. Towards this end, we randomly sample $\{100, 400, 800, 1200, 1600, 2000\}$ images from taskonomy to produce the model transfer-

Figure S5: The model transferability matrix (left) and the task similar tree (right) produced by saliency.

Figure S6: The model transferability matrix (left) and the task similar tree (right) produced by gradient*input.

ability. Results are provided in Figure S4. It can be seen that with the increasing probe data, the constructed relatedness (shown in task similar trees and the visualization of transferability matrices) keeps almost unchanged. This result indicates that the proposed method is also insensitive to the size of the probe data. A few hundreds of images are sufficient for the proposed method.

## 5 Experiments on the Merged Group

To better understand our method, we also merge the two groups (taskonomy models and the pre-trained models collected outside taskonomy) to form a comprehensive group, which consists of totally 38 models. Experiments are conducted on the taskonomy data with saliency, gradient*input and $\epsilon$-LRP, of which results are shown in Figure S5, S6 and S7, respectively. By comparing the results of the three attribution methods, we make the following three observations or analysis:

- $\epsilon$-LRP and saliency produce highly similar results. However, the results of saliency are a little different. These results are consistent with results on taskonomy models or collected models alone, which implies that the proposed method can be scalable to model library of larger size.

- The same-task models, although trained on data from different domains, tend to cluster together. These can be verified by the inpainting models which cluster together in our experiments. Furthermore, we additionally conduct an experiment (not shown in the figure) on another colorization model [12]. The model affinity obtained by our method ranks the taskonomy colorization model first among all others, which again verifies our conclusion.

Figure S7: The model transferability matrix (left) and the task similar tree (right) produced by $\epsilon$-LRP.

- In most cases, the global task structure (all models are considered) preserves the local task structure (only a fraction of models are considered). For example, when removing the taskonomy models from the task similar tree, the remaining tree structure is highly similar with that of collected models.

We argue all these observations are not trivial and providing us more insights into deep models and transfer learning.

## Footnotes

[1]https://github.com/StanfordVL/taskonomy/tree/master/taskbank

[2]https://github.com/tensorflow/models/tree/master/research/slim

[3]https://github.com/JiahuiYu/generative_inpainting