[Reviews · NeurIPS 2019]

Reviewer 1



The authors propose to apply attribution maps to quantify the transferability of deep models, and achieves reasonably accurate results with much faster estimation speed. The approach is, despite simple, quite novel and reasonable. Unlike the taskonomy one that relies on training pairwise or higher-order transfers using annotations, the approach raised here requires no labels and runs very fast. Extensive results demonstrate the validity of the proposed approach. The manuscript is written well; the approach is well motivated, interesting, and practical, as it finds its application in scenarios where pre-trained models are available yet no annotations are provided. The approach could be potentially inspiring to a large audience. ___ After rebuttal After reading all the review comments and the author feedback, I keep my original score. I think this paper makes a good contribution to NeurIPS.

Reviewer 2



The core idea is to measure the task relatedness via the similarity of attribution maps. The paper is overall well written and easy to follow. The method poses barely any constraint on the model architectures, requires no labelled data, and is several-magnitude times faster than taskonomy, which makes it very practical. Furthermore, the fact that it enables flexible insertion of new tasks makes the approach more attractive. Some experimental results are very impressive. It will be better if the authors could show and analyze some bad cases, where two tasks are very related but the corresponding attribution maps are not so similar. The authors should provide more discussions on the rationale behind the fact that, the proposed method works well even the probe data are quite different from the training data of the trained models. In the experiments, SVCCA seems to be working well. What’s the advantage of the proposed method over SVCCA?

Reviewer 3



The submission is well-written and organized in general. It is easy to follow and the motivation is clear. The proposed method is straightforward and technically sound even no source codes are available for justification. Originality: incremental. The proposed investigation aims to address the supervised learning issue in "taskonomy" by employing attribution maps. Quality and clarity: can be further improved. See 5. Significance: limited. See 5.

Reviewer 4



Weaknesses: 1) There are 3 sources for similarities/transferabilities reported in this paper: attribution maps (the proposed method), SVCCA, and Taskonomy. The transferabilities of taskonomy have a practical value (they’re constructed and are shown to reduce the need for supervision through transfer learning), but Taskonomy’s method is computationally expensive. So, the gold standard is duplication of taskonomy’s affinity matrix, but with less complexity. Therefore I see the comparison between the transferability matrix by attribution maps and taskonomy’s (fig 4) valid and what the main point is. But I don’t understand why/how SVCCA vs attribution map’s similarity matrix comparisons (figure 3) are useful. What exactly is the value of SVCCA based similarity matrix? Why isn’t figure 3 comparing between attribution map’s matrix and Taskonomy’s affinity matrix (after being made symmetric)? As I said the practical value of task similarity has been shown for the taskonomy affinity matrix (Fig 7 of Taskonomy paper), so it makes sense to aspire to duplicate that, regardless of SVCCA. In this regard the paper in L225-228 brings in a question to justify comparing against SVCCA: if similarity between attribution maps correlates with similarity between representations by a neural network. But as I reiterated, an absolute similarity between representations of neural networks dont seem to have any practical value unless that similarity is shown to mean transferability (which is what taskonomy affinity matrix does). So why this evaluated assumption is relevant, beyond the comparison with tasjonomy’s affinity matrix, is unclear to me. 2) Related to the above point, the paper seem to suggest attribution maps and SVCCA in the end yield similar task similarity matrices (Fig 3). Then why do the authors believe attribution maps is a novel method that is worth publication, if its final outcome is the same as SVCCA’s? Like the proposed method SVCCA also doesn't need separate labeled data, so supervision is not the advantage. If compute is the advantage, then it should be reported to be clear by how much the attribution maps are more efficient (though I dont find only computational efficiency as an exciting advantage, at least compared to not needing labeled data). Overall, I think the role of SVCCA should be clarified in this submission. 3) The proposed method strictly results in a symmetric task similarity matrix (eq 1 and L174). This seem like a strong constraint and limitation, as the task transferability is not a symmetric property (ie if A transfers well to B, that doesn’t mean B will transfer well to A -- see Fig 7 of taskonomy paper). This makes sense when thinking of task transferability in an information theoretic manner. However, I’m surprised that Fig 4 shows that the symmetric task similarity matrix by attribution maps can be a good prediction of traskonomy’s asymmetric transferability relationships. Are the taskonomy’s top-k sources retrieved as-is from the Taskonomy’s affinity matrix, or are they forced to be symmetric beforehand? Overall, how limiting is the symmetry constraint (ideally reported quantitatively). 4) I didn’t quite find the attribution maps qualitatively intuitive (Fig 2). The attended areas of the image don’t seem to be related to the actual task (e.g. in 3D tasks the 3D worth pixels don't seem to be attended). Or there seem to be some clusters of attended pixels without a clear semantic meaning behind them. However, the resulting analysis using the attribution maps seem to work (sec 4.2), so quantitative value seem to exist. But as this state I fail to spot a qualitative value. 5) related to point 1 above, the analysis in sec 4.3 is more like a curious experiment and intuitive evaluations of the trends. As SVCCA (ie just similarity between representations) doesn’t mean transferability value necessarily, the trends in section 4.3 do not necessarily have a practical value. However, I still think having them is better than removing them, but I would clarify the observed trends don’t necessarily have a conclusive practical value. 6) Per Fig 4, it seems that the probe datasets other than taskonomy better predict the taskonomy transferability than the probe dataset based on taskonomy itself. This seems counter intuitive. How do you justify this? 7) The paper should cite and compare with the recent works that also attempt to duplicate taskonomy’s affinity matrix but with cheaper methods. E.g. “Representation Similarity Analysis for Efficient Task taxonomy & Transfer Learning”, CVPR19.

[Author Response · NeurIPS 2019]

**Rebuttal: Deep Model Transferability from Attribution Maps** (Paper ID 3328)

We would like to thank the AC and all the reviewers for the constructive comments, and would like to address them
as follows. Due to the page limit, we provide short responses but will include more details in the final version.
———————————————————————— **To Reviewer #1** ————————————————————————
**Q1:** How the performance will be affected if the attribution maps are quantified?
**A1:** We evaluate the proposed method using different number of bits, $\{1, 2, 4, 8, 16, 32\}$, to represent an element in the
attribution maps. Spearman correlations between the result (affinity matrix) of 32 bits and those of $\{1, 2, 4, 8, 16\}$ bits
are $\{0.56, 0.71, 0.85, 0.96, 0.99\}$, respectively. It can be seen that, with appropriately fewer bits the proposed method
also works well; however, too few (1 or 2) bits may largely affect the result.
**Q2:** What is the principle for choosing the layer for computing attribution maps?
**A2:** All taskonomy models follow the encoder-decoder architecture. For these models, we choose the output of the
encoder to compute the attribution maps. Non-taskonomy models, in fact, can also be viewed as encoder-decoder ones.
For example, in classification models, the convolution layers can be viewed as the encoder and the fully connected
layers as the decoder. The attribution maps are thus also computed with respect to the output of the encoder.
**Q3:** How the attribution maps change as the layers go deeper? Please provide more results in the supplement material.
**A3:** Thanks. Shallow layers produce attribution maps where relevance scores are distributed uniformly; as the layers go
deeper, the attribution maps tend to focus more on task-relevant regions. More details will be added to the revision.
———————————————————————— **To Reviewer #2** ————————————————————————
**Q1:** It will be better if the authors could show and analyze some bad cases.
**A1:** Thanks for the comment. For the target task Class Places, the obtained order of source tasks from our method is in
fact not so similar to that produced by taskonomy. The main reason may be that, in taskonomy, most models are trained
for 2D, 3D, or low dimensional geometric tasks that are very different from Class Places. These tasks may produce
comparable performances when transferred to Class Places, so that the rank of source tasks is not really meaningful.
**Q2:** The authors should provide more discussions on the rationale behind the fact that, the proposed method works well
even the probe data are quite different from the training data of the trained models.
**A2:** Thanks. Our basic assumption is, trained models of similar tasks should produce similar attribution maps given the
same data that are randomly sampled, even if these data are from a different domain from the training data. We will
provide more discussion on this issue in the final version.
**Q3:** What's the advantage of the proposed method over SVCCA?
**A3:** The main advantage is efficiency. In SVCCA, we need to compute the correlation between the features of every
two tasks. However, in our method, we only need to project the pre-trained models into a common model space. The
task affinity matrix is derived from the distance between points in this space, in a plug-and-play fashion.
———————————————————————— **To Reviewer #4** ————————————————————————
**Q1:** The usage of mathematical symbols should be consistent.
**A1:** Thanks for pointing out this issue. We will revise the inconsistencies in the final version.
**Q2:** Experimental mistake? In Figure 4, according to the precision and recall curves (the higher the better), saliency is
better than $\epsilon$-LRP and gradient. However, in Line 267, a completely reversed conclusion is given.
**A2:** Thanks for the comments. There is indeed no mistake here. We believe the reviewer might have taken the *oracle*
curve in Fig. 4 for the *taskonomy_saliency* curve, both of which have a similar color. In fact, the three saliency curves
(*taskonomy_saliency*, *indoor_saliency* and *coco_saliency*) are lower than other curves except that of *random ranking*,
meaning that the results are consistent with our conclusion. We will tune the curve colors to avoid such confusion.
**Q3:** Since all the three attribution methods are employed from previous work, it would be better to present more
discussions on the relation/interpretation/understanding among Saliency, Gradient*Input, and $\epsilon$-LRP maps.
**A3:** Thanks for the suggestion. In short, Saliency constructs attributions by taking the absolute value of the partial
derivative of the target output with respect to the input. Gradient*Input refers to a first-order Taylor approximation
of how the output would change if the input was set to zero. $\epsilon$-LRP, on the other hand, computes the attributions by
redistributing the prediction score (output) layer by layer until the input layer is reached. As suggested, we will provide
more details of the three in the final version to make the paper easier to follow.
**Q4:** Some conclusions are kind of trivial. For instance, "all ImageNet-trained models tend to cluster together", "the
same-task trained models with similar architectures tend to be more related than with dissimilar architectures", and etc.
These conclusions are rather obvious since the model embeddings are calculated using gradients w.r.t. input images.
**A4:** Thanks for the comment. We would like the remind the reviewer that, despite all the model embeddings are
computed using gradients with respect to input images, we allow the the model architectures, initializations and training
processes to be different. For the same task, therefore, such different configurations may lead to different decision
patterns and hence attribution maps focusing on different regions. Without the experiments we conducted, in our
opinion, it might not be perfectly safe to draw the aforementioned conclusions.
**Q5:** No source code is provided.
**A5:** We promise that the source code, data, and models will be released for reproducing the results in the paper.

[Meta-Review · NeurIPS 2019]

The contribution allows to compare heterogeneous networks by projecting them in a model space based on attribution maps (a task & model dependent attention map over inputs). The distance between the embedding of the networks in the model space is used as good metric for measuring task transferability in a very simple and cheap way. This is interesting and allows to provide good estimators of transfer capability. The justification of some choices could be improved with the help of the reviews.